# Inhibition of Amyloid-β (Aβ)-Induced Cognitive Impairment and Neuroinflammation in CHI3L1 Knockout Mice through Downregulation of ERK-PTX3 Pathway

**DOI:** 10.3390/ijms25105550

**Published:** 2024-05-19

**Authors:** Hyeon Joo Ham, Yong Sun Lee, Ja Keun Koo, Jaesuk Yun, Dong Ju Son, Sang-Bae Han, Jin Tae Hong

**Affiliations:** College of Pharmacy and Medical Research Center, Chungbuk National University, Osongsaengmyeong 1-ro, Osong-eup, Heungdeok-gu, Cheongju 28160, Republic of Korea

**Keywords:** Alzheimer’s disease, neuroinflammation, CHI3L1, NF-κB, ERK, PTX3

## Abstract

Several clinical studies reported that the elevated expression of Chitinase-3-like 1 (CHI3L1) was observed in patients suffering from a wide range of diseases: cancer, metabolic, and neurological diseases. However, the role of CHI3L1 in AD is still unclear. Our previous study demonstrated that 2-({3-[2-(1-Cyclohexen-1-yl)ethyl]-6,7-dimethoxy-4-oxo-3,4-dihydro-2-quinazolinyl}culfanyl)-*N*-(4-ethylphenyl)butanamide, a CHI3L1 inhibiting compound, alleviates memory and cognitive impairment and inhibits neuroinflammation in AD mouse models. In this study, we studied the detailed correlation of CHI3L1 and AD using serum from AD patients and using CHI3L1 knockout (KO) mice with Aβ infusion (300 pmol/day, 14 days). Serum levels of CHI3L1 were significantly elevated in patients with AD compared to normal subjects, and receiver operating characteristic (ROC) analysis data based on serum analysis suggested that CHI3L1 could be a significant diagnostic reference for AD. To reveal the role of CHI3L1 in AD, we investigated the CHI3L1 deficiency effect on memory impairment in Aβ-infused mice and microglial BV-2 cells. In CHI3L1 KO mice, Aβ infusion resulted in lower levels of memory dysfunction and neuroinflammation compared to that of WT mice. CHI3L1 deficiency selectively inhibited phosphorylation of ERK and IκB as well as inhibition of neuroinflammation-related factors in vivo and in vitro. On the other hand, treatment with recombinant CHI3L1 increased neuroinflammation-related factors and promoted phosphorylation of IκB except for ERK in vitro. Web-based gene network analysis and our results showed that CHI3L1 is closely correlated with PTX3. Moreover, in AD patients, we found that serum levels of PTX3 were correlated with serum levels of CHI3L1 by Spearman correlation analysis. These results suggest that CHI3L1 deficiency could inhibit AD development by blocking the ERK-dependent PTX3 pathway.

## 1. Introduction

Alzheimer’s disease (AD) is the most common type of neurodegenerative disease that afflicts millions of people worldwide, but efforts to overcome AD over the past decades have failed to achieve the desired clinical efficacy [1]. Through pathological examination of the brain tissue of AD patients, two characteristic proteins, amyloid plaques and phosphate-tau neurofibrillary tangles, are abnormally accumulated, and it is believed that these cause synapse loss and memory impairment [2,3]. Many therapies targeting them have not been shown to be significantly effective in treating AD and, more recently, have moved toward targeting other factors and processes involved in the development of AD [4]. Evidence is accumulating to show that inflammatory processes (neuroinflammation) that occur within the central nervous system (CNS) are not merely pathological features but may in themselves be causative [5,6]. Neuroinflammation is important for neuronal and brain homeostasis, but when chronic, it turns microglia and astrocytes into a reactive toxic state in which they can release molecules that damage neurons, contributing to the pathogenesis of neurodegenerative diseases, including AD [7]. Microglial activation is a part of neuroinflammation, and recently, it has been known that microglia are polarized into two phenotypes, M1 (traditional activation) phenotype and M2 (alternative activation) phenotype in response to different micro-environmental disturbances [8]. The M1 phenotype releases inflammatory cytokines and chemokines resulting in neuronal cell death, while the M2 phenotype releases anti-inflammatory cytokines and chemokines resulting in neuroprotection, both of which are involved in the pathogenesis of AD [8,9].

The chitinase-like protein family is a glycoprotein similar to chitinase, a plant hydrolase that catalyzes the breakdown of chitin to protect the host against various pathogens and lacks enzymatic activity [10,11]. They are emerging as biomarkers for a wide range of neurological disorders, but their physiological role remains unclear [12]. First discovered in human osteosarcoma cells, Chitinase-3-like 1 (CHI3L1), also called YKL-40, is the most widely known and studied chitinase-like protein [13]. CHI3L1 is expressed in several types of cell types, including neutrophils, macrophages, chondrocytes, and fibroblast-like synovial cells [11,14]. CHI3L1 has been widely studied as it has been found to be significantly involved in various diseases in the peripheral system, including asthma, arthritis, sepsis, diabetes, liver fibrosis, coronary artery disease, and several cancers, but its role in brain health and disease has not been investigated much even though it is expressed not only in peripheral systems but also in the CNS such as microglia and astrocytes [14,15,16,17]. Overexpression of CHI3L1, produced by cells (neutrophils, macrophages, fibroblast-like cells, T-lymphocytes, and endothelial cells) involved in chronic inflammation, is considered a hallmark of various inflammatory conditions [18,19]. Several studies have reported that CHI3L1 is involved in the inflammatory response such as the secretion of pro-inflammatory cytokines, showing that excessive levels of CHI3L1 can initiate and sustain chronic inflammation [20,21]. Elevated levels of CHI3L1 have been reported in the cerebrospinal fluid of patients with AD compared to normal subjects in a progression-dependent manner [22,23]. In addition, as shown in the study, the expression of pro-inflammatory cytokines, including tumor necrosis factor (TNF)-α, interleukin (IL)-1β, and IL-6, was reduced through silencing CHI3L1 in mice, and in the studies on the association between the extracellular signal-regulated kinases (ERK) or nuclear factor-kappa B (NF-κB) signaling pathway and CHI3L1, there is growing evidence that CHI3L1 plays a major role in the inflammatory responses [24,25,26,27]. However, little is known about the role of CHI3L1 in neuroinflammation and AD pathology, including microglia polarization.

In our previous studies, we found that the CHI3L1 inhibitor, 2-({3-[2-(1-cyclohexen-1-yl)ethyl]-6,7-dimethoxy-4-oxo-3,4-dihydro-2-quinazolinyl}sulfanyl)-*N*-(4-ethylphenyl)butanamide (K284-6111), alleviated memory dysfunction and neuroinflammation in β-amyloid (Aβ-induced) and transgenic (Tg2576) AD mouse model [11,27]. In this study, building on evidence found by other researchers and our previous studies, the relationship between CHI3L1 and AD pathology was studied in more detail using serum from AD patients and using CHI3L1 knockout (KO) mice-induced AD.

## 2. Results

### 2.1. CHI3L1 Is Related to AD in the Serum of AD Patients

CHI3L1 is upregulated in many types of cancer and in diseases with chronic inflammation including rheumatoid arthritis, osteoarthritis, liver fibrosis, inflammatory bowel disease, and bacterial sepsis [28]. We performed an ELISA assay to investigate the clinical association between CHI3L1 and AD. We quantitatively measured the levels of Aβ_1–40_, Aβ_1–42_, and Tau, the markers of AD according to the amyloid cascade and Tau hypothesis, and CHI3L1 in the serum of AD patients and healthy controls by ELISA. The serum levels of Aβ_1–40_ in AD patients were significantly increased (0.16 ± 0.02 ng/mL) compared to that of healthy controls (0.11 ± 0.01 ng/mL) (*n* = 25, *p* = 0.0105). The serum levels of Aβ_1–42_ in AD patients were significantly increased (0.47 ± 0.06 ng/mL) compared to that of healthy controls (0.56 ± 0.04 ng/mL) (*n* = 25, *p* = 0.2001). The serum levels of Tau in AD patients were also significantly increased (554.5 ± 27.0 ng/mL) compared to that of healthy controls (489.4 ± 10.6 ng/mL) (*n* = 25, *p* = 0.0294). Consistent with the significant elevation of Aβ_1–42_, the serum levels of CHI3L1 in AD patients were significantly elevated (29.5 ± 0.7 ng/mL) compared to that of healthy controls (16.2 ± 0.7 ng/mL) (*n* = 25, *p* < 0.001) (Figure 1A).

Receiver operating characteristic (ROC) analysis is a widely used method in clinical epidemiology to quantify how accurately a medical diagnostic test could distinguish a disease state from a non-disease state [29]. Thus, we performed ROC analysis for these four proteins, Aβ_1–40_, Aβ_1–42_, Tau, and CHI3L1. The area under the curve (AUC), AD diagnosis cut-off value, specificity, and sensitivity of Aβ_1–40_ were 0.7000, 0.1286, 0.84, and 0.56, respectively. The AUC, AD diagnosis cut-off value, specificity, and sensitivity of Aβ_1–42_ were 0.6704, 0.4430, 0.65, and 0.72, respectively. The AUC, AD diagnosis cut-off value, specificity, and sensitivity of Tau were 0.6096, 545.6, 0.84, and 0.44, respectively. The AUC, AD diagnosis cut-off value, specificity, and sensitivity of CHI3L1 were 0.9856, 25.56, 1, and 0.88, respectively. The ROC analysis indicated that CHI3L1 is a good candidate for an AD biomarker compared to the other markers (Figure 1B). We conducted the Spearman’s correlation test to investigate whether CHI3L1 and AD markers, Aβ_1–40_, Aβ_1–42_, and Tau, are correlated. The Spearman’s correlation showed that there is high correlation between CHI3L1 and Aβ_1–40_, but there is no high correlation between CHI3L1 and Aβ_1–42_ or Tau (*n* = 25, Aβ_1–40_: *r^2^* = 0.2785, *p* = 0.0067; Aβ_1–42_: *r^2^* = 0.1124, *p* = 0.1014; Tau: *r^2^* = 0.2097, *p* = 0.0213) (Figure 1C). These data indicate that CHI3L1 could be an independent marker for AD.

### 2.2. CHI3L1 Deficiency Suppresses Memory Impairment Induced by Aβ

To investigate the role of CHI3L1 in AD, CH3L1 KO and WT were injected with oligomeric Aβ into the ventricles to mimic the neuropathological features of late-onset AD. To explore the effect of CHI3L1 on memory impairment, which is one of the most characteristic features of AD, oligomeric Aβ (300 pmol/day) was infused to CH3L1 KO and WT mice for 14 days. After 14 days, a series of behavioral tests were performed to assess the learning ability and memory of the mice (Figure 2A). The water maze test to assess spatial learning and memory abilities, the probe test to assess the degree of memory enhancement, and the passive avoidance test to assess memory retention abilities were sequentially performed.

In the water maze test, vehicle-treated WT and KO mice showed no significant difference in mean escape latency and swimming distance for 6 days, but Aβ-infused WT and KO mice showed a significant difference from day 5 of the water maze test. On the final day, the Aβ-infused WT mice showed significantly increased mean escape latency and swimming distance compared to that of the control WT mice, which were 37.9 ± 6.1 s and 3206 ± 404.9 cm, respectively (*n* = 8–10, *p* < 0.001 and *p* = 0.0017, respectively). The Aβ-infused KO mice showed significantly decreased mean escape latency and swimming distance compared to that of the Aβ-infused WT mice, which were 17.1 ± 3.5 s and 2137 ± 132.0 cm, respectively (*n* = 8–10, *p* = 0.0103 and *p* = 0.0319, respectively; Figure 2B,C).

On the day after day 6 of the water maze test, the platform that was covered with water was removed and then the probe test was performed. In the probe test, the percentage of the mean time spent in the target quadrant, where the platform was located as an indicator of memory consolidation, was expressed. The mean time spent in the target quadrant was significantly different between WT and KO controls (35.2 ± 2.6% and 38.6 ± 2.3%, respectively; *n* = 8–10, *p* = 0.3596). On the other hand, the Aβ-infused WT group showed significantly lower mean times compared to that of WT controls (16.8 ± 0.7%; *n* = 8–10, *p* < 0.0001), and the Aβ-infused KO group showed significantly higher mean times than that of Aβ-infused WT group (27.2 ± 2.1%; *n* = 8–10, *p* = 0.0024; Figure 2D).

In the passive avoidance test, there was no significant difference between all groups in the training trial. In the test trial, the Aβ-infused WT group showed a significantly lower mean step-through latency compared to that of WT control group (55.0 ± 9.0 s and 201.2 ± 30.6 s, respectively; *n* = 8–10, *p* < 0.0001), and the Aβ-infused KO group showed a significantly high mean step-through latency compared to that of Aβ-infused WT group (89.2 ± 10.0 s; *n* = 8–10, *p* = 0.0241; Figure 2E). Taken together, there was significantly less impairment of Aβ-induced cognitive impairment in CHI3L1 KO mice compared to WT mice.

### 2.3. CHI3L1 Deficiency Inhibits Aβ Deposition in Aβ-Induced AD Mouse Brain

The most accepted hypothesis for the pathogenesis of AD is the amyloid cascade, in which oligomerized Aβ-peptides at an early stage play a central role in the pathology. To investigate the effect of the absence of CHI3L1 on the deposition of oligomeric Aβ plaques in the mouse brain, thioflavin S staining was performed to stain β-sheet-rich structures of oligomeric Aβ plaques. Aβ plaques were not significantly different in WT and KO controls but were more present in the Aβ-infused WT mice than in the WT controls. However, fewer Aβ plaques were observed in Aβ-infused KO mice than in Aβ-infused WT mice (Appendix A). ELISA was performed to quantitatively compare the amount of Aβ in the brains of each group of mice. The Aβ_1–40_ level in the mouse hippocampus was significantly elevated in Aβ-infused WT mice (15.6 ± 0.8 μg/mg of protein) compared to that of WT control mice (1.5 ± 0.1 μg/mg of protein). In addition, a significantly lower concentration of Aβ_1–40_ was detected in the hippocampus of Aβ-infused KO mice (11.4 ± 0.8 μg/mg of protein) compared to that of Aβ-infused WT mice (*n* = 8–10, F(3, 28) = 103.10, *p* < 0.0001; Appendix A). The Aβ_1–42_ level in the mouse hippocampus was significantly elevated in Aβ-infused WT mice (2.3 ± 0.1 μg/mg of protein) compared to that of WT control mice (0.2 ± 0.04 μg/mg of protein). In addition, a significantly lower concentration of Aβ_1–42_ was detected in the hippocampus of Aβ-infused KO mice (1.3 ± 0.2 μg/mg of protein) compared to that of Aβ-infused WT mice (*n* = 8–10, F(3, 28) = 103.10, *p* < 0.0001; Appendix A). Given these data, it is suggested that Aβ deposition in the brain of mice was reduced due to the absence of CHI3L1.

### 2.4. The Absence of CHI3L1 Suppresses Neuroinflammation Induced by Aβ in Mouse Brain

Over the past decade, it has been known that the sustained activation of microglia and other cells involved in immune responses in the brain may play roles in the pathogenesis of AD [30]. To investigate the effect of the absence of CHI3L1 on neuroinflammation, immunohistochemistry, qRT-PCR, and Western blot were performed to compare the expression and levels of factors related to neuroinflammation in four mouse groups. Aβ-induced reactive astrocyte and activated microglia and inflammatory marker proteins (iNOS and COX-2) were decreased in brain tissue of Aβ-infused KO mice compared to those in Aβ-infused WT mice (Figure 3A,D). The Aβ-induced activation of ERK and NF-κB signaling was reduced in the brain of Aβ-infused KO mice compared to those in Aβ-infused WT mice (Figure 3E). In addition to these results, our qRT-PCR results showed that elevated expression of inflammatory cytokines and chemokines (*Tnf*, *Il1b*, *Il6*, and *Ccl2*) and the markers of M1 microglia phenotype (*Cd16*, *Cd32*, *Cd68*, and *Cd86*) were reduced in the brain of Aβ-infused KO mice compared to those in Aβ-infused WT mice (*n* = 10–12; *Tnf*: F(3, 27) = 25.95, *p* < 0.0001; *Il1b*: F(3, 28) = 73.31, *p* < 0.0001; *Il6*: F(3, 28) = 40.74, *p* < 0.0001; *Ccl2*: F(3, 28) = 42.20, *p* < 0.0001; *Cd16*: F(3, 25) = 31.85, *p* < 0.0001; *Cd32*: F(3, 27) = 71.10, *p* < 0.0001; *Cd68*: F(3, 28) = 53.45, *p* < 0.0001; *Cd86*: F(3, 28) = 67.97, *p* < 0.0001) (Figure 3B,C).

### 2.5. CHI3L1 Is Associated with Inflammatory Responses in BV-2 Microglia Cell

In our previous studies, using K284-6111, which exerts an inhibitory effect by direct binding to CHI3L1, we demonstrated that CHI3L1 inhibition alleviated the LPS and Aβ-induced neuroinflammatory responses in BV-2 cells [11,27]. But it has not been demonstrated whether direct blocking of CHI3L1 expression could suppress the induction of neuroinflammation. Therefore, we investigated whether CHI3L1 knockdown attenuates inflammation in BV-2 microglial cells. The deficiency of CHI3L1 using small interfering RNA (siRNA) decreased expressions of protein involving inflammation, iNOS, COX-2, and IBA-1, and decreased the activation of ERK and NF-κB in Aβ-induced BV-2 cells (Figure 4A,B). The CHI3L1 knockdown BV-2 cells showed lower mRNA expression levels of pro-inflammatory cytokines and chemokines (*Tnf*, *Il1b*, *Il6*, and *Ccl2*) (*n* = 12; *Tnf*: F(3, 44) = 33.32, *p* < 0.0001; *Il1b*: F(3, 44) = 19.66, *p* < 0.0001; *Il6*: F(3, 44) = 125.9, *p* < 0.0001; *Ccl2*: F(3, 44) = 54.61, *p* < 0.0001) and M1 microglia markers (*Cd16*, *Cd32*, *Cd68*, and *Cd86*) (*n* = 10–12; *Cd16*: F(3, 41) = 33.32, *p* = 0.0003; *Cd32*: F(3, 44) = 16.12, *p* < 0.0001; *Cd68*: F(3, 44) = 202.0, *p* < 0.0001; *Cd86*: F(3, 44) = 98.82, *p* < 0.0001) (Figure 4C,D). To elucidate whether CHI3L1 could induce neuroinflammation in microglia, BV-2 cells were treated with rmCHI3L1. rmCHI3L1 treatment induced expressions of iNOS, COX-2, and IBA-1 and induced the activation of ERK and NF-κB in BV-2 cells (Figure 5A,B). The mRNA pro-inflammatory cytokines and chemokine, and M1 microglia markers, excluding *Tnf* and *Cd68*, were significantly elevated by rmCHI3L1 treatment in BV-2 cells (*n* = 11–12; *Tnf*: *p* = 0.5001; *Il1b*: *p* = 0.0063; *Il6*: *p* = 0.0341; *Ccl2*: *p* = 0.0002; *Cd16*: *p* = 0.0019; *Cd32*: *p* = 0.0041; *Cd68*: *p* = 0.1902; *Cd86*: *p* = 0.0143) (Figure 5C,D).

### 2.6. CHI3L1 Is Related to PTX3

In our previous study, we presented that CHI3L1 may be related to PTX3 by means of the web-based GWAS analysis and demonstrated the relationship between CHI3L1 and PTX3 and the relationship between PTX3 and neuroinflammation using a transgenic AD mouse model and Aβ-induced BV-2 cells (Figure 6A) [11]. In the present study, we showed that CHI3L1 significantly increased in the serum of AD patients; thus, we measured PTX3 levels in the serum of AD patients and healthy controls using ELISA. The serum levels of PTX3 in AD patients were significantly increased (5.8 ± 0.6 ng/mL) compared to that of healthy controls (3.5 ± 0.4 ng/mL) (*n* = 25, *p* = 0.0011) (Figure 6B). To quantify how PTX3 accurately could distinguish AD from healthy controls, we carried out ROC analysis for PTX3. The AUC, AD diagnosis cut-off value, specificity, and sensitivity of PTX3 were 0.7856, 3.426, 0.64, and 0.96, respectively (Figure 6C). In order to examine the correlation between CHI3L1 and PTX3, we conducted the Spearman’s correlation test. The Spearman’s correlation showed that the serum level of PTX3 correlated with the serum level of CHI3L1 (*n* = 25, *r^2^* = 0.1786, *p* = 0.0353) (Figure 6D). To investigate the effect of the absence of CHI3L1 on PTX3 in the brain, a Western blot was carried out to compare the PTX3 level in four mouse groups. Levels of PTX3 were lower in KO control mice compared to WT control mice, increased by Aβ infusion, and slightly lower in Aβ-infused KO mice than Aβ-infused WT mice (Figure 6E).

## 3. Discussion

As a significantly high concentration of CHI3L1 is observed in the cerebrospinal fluid of Alzheimer’s patients, CHI3L1 could be considered as a biomarker for various diseases involving chronic inflammatory responses including AD [12,18,19,22,23]. Also, we have reported in our previous studies that inhibition of CHI3L1 alleviates impaired cognitive function and memory in the Tg2576 AD mouse model and LPS-induced AD mouse model using K284-6111, which functions to directly bind and inhibit CHI3L1 [11,27]. These data suggest that CHI3L1 may be significantly involved in AD, but the detailed mechanisms of CHI3L1 in AD development remain unclear. In this study, we induced AD pathology through Aβ infusion to the brain in CHI3L1 WT mice and CHI3L1 KO mice and investigated the effect of CHI3L1 on AD and neuroinflammation. We demonstrated that CHI3L1 deficiency attenuates the development of Aβ-induced memory impairment and neuroinflammation in mice. Deficiency of CHI3L1 inhibits the activation of NF-κB and ERK-PTX3 signaling, thereby inhibiting inflammatory factors and M1 polarization of microglia.

There are various contributing factors to the pathogenesis of AD, and the main features of AD include loss of synapses and neurons, extracellular plaques and intracellular neurofibrillary tangles Aβ, and hyperphosphorylated tau, respectively [31]. It is widely accepted that Aβ activates microglia and triggers the release of pro-inflammatory cytokines, which induces the production of amyloid precursor protein (APP), and that increased APP leads to increased Aβ production [32]. Neuroinflammation, another pathological feature of AD, is accompanied by activation of microglia and astrocytes, and inflammatory responses such as pro-inflammatory cytokines and reactive oxygen species released through their activation are toxic to neurons, resulting in synaptic defects and neuronal loss [33,34,35]. In this regard, altered levels of several cytokines and chemokines, such as higher pro-inflammatory cytokines including TNF-α, IL-1β and IL-6, and inflammatory chemokines including CCL2, have been reported in the body fluids of AD patients compared to normal subjects [36,37,38]. In the CNS, CHI3L1 is most highly expressed in astrocytes and relatively less expressed in other microglia and neurons, so studies of CHI3L1 in the CNS mainly focus on its role in astrocytes [3]. However, in our study, CHI3L1 knockout not only lowered the activation of astrocytes but also microglia. We showed that the activation of microglia and astrocytes was increased in the hippocampus of AD-induced mice with Aβ infusion by comparing the expressions of IBA-1, a marker of activated microglia, and GFAP, a marker of activated astrocytes, but not CHI31 KO mice hippocampus. In CHI3L1 KO mice, the activation of cells involved in two major immune responses in the CNS was lowered, and the expression of pro-inflammatory markers, such as pro-inflammatory cytokines (*Tnf*, *Il1b*, and *Il6*) and chemokines (*Ccl2*), and inflammatory proteins (iNOS and COX-2), was also lowered. Microglia were classified into pro-inflammatory M1 phenotype and anti-inflammatory M2 phenotype, like the classification of macrophage phenotype into M1 and M2 [39]. In the hippocampus of AD-induced CHI3L1 KO mice, the expression level of the M1 markers, *Cd16*, *Cd32*, *Cd68*, and *Cd86*, was lower than that of WT; therefore, CHI3L1 KO inhibited the polarization toward the M1 phenotype. Interestingly, no significant changes were observed in the expression of markers associated with inflammation (cytokines, chemokines, proteins, and M1 markers) in the hippocampus of WT mice and CHI3L1 KO mice without AD induction. In the same context, the composition of the CHI3L1-deficient environment by CHI3L1 siRNA reduced pro-inflammatory cytokines, and the markers of M1 phenotype, and inflammatory proteins. However, the composition of the CHI3L1-rich environment and treatment of rmCHI3L1 increased these pro-inflammatory molecules in BV-2 cells. Moreover, CHI3L1 induced the increase in expressions of iNOS and COX-2, and the knockdown of CHI3L1 offsets these increased expressions. These data suggest that CHI3L1 was involved in the pathogenesis of AD by exaggerating M1-dependent neuroinflammation.

Consistent with our previous studies [11,27], the activation of ERK and NF-κB signals in the hippocampus of CHI3L1 KO mice was lower than that of WT. Our in vitro results also showed that treatment of CHI3L1 siRNA in BV-2 cells reduced the Aβ-induced p-ERK and p-IκBα, whereas rmCHI3L1 treatment conversely increased p-ERK and p-IκBα. Currently, a total of six CHI3L1 receptors have been reported: IL-13 receptor alpha 2, transmembrane protein 219, galectin-3, CD44, chemo-attractant receptor-homologous 2, and receptor for advanced glycation end products (RAGE) [14,40,41,42,43]. When CHI3L1 binds to RAGE on the cell surface, it activates NF-κB, MAPK/ERK, and STAT3 signals [40]. It is well known that ligand binding to RAGE activates multiple cellular signals, Ras/MEK/ERK, SAPK/JNK, MAPK/P38, PI3K/AKT, and JAK/STAT, followed by activation of transcription factors, NF-κB, STAT3, AP-1, and Egr-1, to synthesize and release inflammatory cytokines including IL-1, IL-6, and TNF-α [44]. RAGE is highly expressed in neurons and microglia of AD patients, which contributes to the accumulation of Aβ and the formation of neuroinflammatory plaques, accelerating the progression of AD [45,46]. Results from our previous study on the combined effects of CHI3L1 inhibitors and ERK and NF-κB inhibitors show that NF-κB signaling is affected by ERK signaling [11]. Taken together, CHI3L1 mediates neuroinflammation through ERK and NF-κB signaling. 

Moreover, recent studies have shown increased levels of CHI3L1 in CSF in AD patients compared to healthy controls [47]. To investigate what is the significant and suitable biomarker for AD, we studied ROC analysis with Aβ_1–40_, Aβ_1_–_42_, Tau, CHI3L1, and PTX3. We demonstrated that CHI3L1 showed higher AUC values compared to others in the ROC analysis using serum of AD patients and healthy controls, presenting the possibility that CHI3L1 could be used as one of the biomarkers for AD diagnosis. CHI3L1 has the potential to be used as a blood-based biomarker of AD and thus could be a simple and cost-effective biomarker. After confirming the possibility that PTX3 is associated with CHI3L1 through web-based GWAS analysis in our previous study [11], we experimentally demonstrated that PTX3 is regulated by CHI3L1 in BV-2 cells. In addition, Rajkovic et al. reported that chronic CNS disorders including Parkinson’s disease and AD showed increased plasma PTX3 levels clinically and experimentally [48]. These data indicated that the PTX3 pathway could play a significant role in CHI3L1 involved in AD development.

Combining our previous and newly obtained results, these data demonstrate that CHI3L1 is involved in the activation of NF-κB and ERK pathways related to neuroinflammation and that CHI3L1 deficiency could slow AD onset and progression. Within the framework of this interpretation, the development of therapies targeting CHI3L1 could be as effective as AD therapy.

## 4. Materials and Methods

### 4.1. Human Biospecimens

The human biospecimens with AD and healthy controls (25 samples from each group) were provided by the Biobank of Gyeongsang National University Hospital and the Biobank of Chungbuk National University Hospital, the members of Korea Biobank Network. All studies using human biospecimens were conducted in accordance with the Declaration of Helsinki and were approved by the Ethics Committee of Chungbuk National University Medical Center (institutional review board approval No. CBNU-201909-BR-925-01).

### 4.2. ELISA Assay

Human serum samples were analyzed by using ELISA kits purchased from R&D Systems (CHI3L1; Minneapolis, MN, USA), myBioSource (Aβ_1–40_ and Aβ_1–42_; San Diego, CA, USA), and Abcam (Pentraxin-3 (PTX3); Cambridge, MA, USA) following the manufacturer’s instructions. Aβ_1–42_ and Aβ_1–40_ levels were determined using each specific mouse Aβ_1–42_ enzyme-linked immunosorbent assay (ELISA) Kit and mouse Aβ_1–40_ ELISA Kit purchased from CUSABIO (Houston, TX, USA) following the manufacturer’s protocol. 

### 4.3. Materials

The Aβ_1–42_ was purchased from Sigma Aldrich (St. Louis, MO, USA). The recombinant mouse CHI3L1(rmCHI3L1) protein was purchased from R&D Systems (Minneapolis, MN, USA). The rmCHI3L1 was dissolved in PBS (final concentration of 0.1 mg/mL) and stored at −20 °C until use.

### 4.4. Animals

All experimental protocols involving animals were performed in compliance with the guidelines for animal experimentation of the Institutional Animal Care and Use Committee of the Laboratory Animal Research Center at Chungbuk National University, Korea (ethical approval number: CBNUR-1424-20). We made every effort to minimize animal suffering during experiments and reduce the number of animals used. All mice were housed in cages with automatic temperature control (21–25 °C), relative humidity (45–65%), and 12 h light–dark cycle illumination, with a maximum of three mice assigned to one cage. Filtered tap water and a rodent chow diet were provided ad libitum throughout the experiment. Wild-type (WT) (C57BL/6) mice were purchased from DBL (Eumsung, Republic of Korea). CHI3L1 knockout (KO) mice were generated as described in a previous study [49].

### 4.5. Preparation of Oligomeric Aβ_1–42_

Oligomeric Aβ_1–42_ was prepared as described in the publication of Jean et al. [50]. The Aβ_1–42_ peptide was dissolved in dimethyl sulfoxide (DMSO; final concentration of 5 mM) and stored at −20 °C until use. Two days before surgery, 5 mM Aβ_1–42_ solution was diluted to 100 μM with sterile saline and incubated at 4 °C for 12 h. The day before surgery, the incubated 100 μM Aβ_1–42_ solution was diluted to 50 µM with sterile saline. It was then filled into an Osmotic pump 1002 model coupled with Brain Infusion Kit 3 purchased from ALZET (Cupertino, CA, USA) and primed overnight according to the manufacturer’s instructions.

### 4.6. Aβ_1–42_-Infused Mouse Model

Age-matched WT mice and CHI3L1 KO mice (10-week-old) in the C57BL/6 background strain were used. WT mice and CHI3L1 KO mice were randomly divided into two groups: (Ⅰ) the control vehicle-treated group (*n* = 10) and (Ⅱ) the Aβ_1–42_ (300 pmol/day)-induced group (*n* = 10). To induce the neuronal death aspect of amyloidopathy, 300 pmol of oligomeric Aβ_1–42_ was directly infused into the mouse brain for 14 days via implantation of a mini-osmotic pump. Control mice were alternatively given an Aβ_1–42_-deficient vehicle of the same composition. Under isoflurane anesthesia, the cannula was implanted in the dentate gyrus of the hippocampus (AP, −2.00 mm; ML, ±1.3 mm; DV, −2.2 mm), and the mini-osmotic pump was implanted under the skin in the interscapular region. After surgery, the antibiotic ointment was applied to the incision site once every 2 days, and the behavioral tests assessing learning and memory capacity (the water maze, probe, and passive avoidance tests) were performed on mice after 14 days of recovery and induction period. Mice were sacrificed after behavioral tests by CO_2_ asphyxiation.

### 4.7. Morris Water Maze

The water maze test is a commonly accepted method for assessing cognitive function, and we performed it as described by Morris et al. [51]. We filled a circular plastic pool (height: 35 cm, diameter: 100 cm) with opaque white water prepared using skim milk and maintained at a temperature of 22–25 °C.

An escape platform (height: 14.5 cm, diameter: 4.5 cm) was placed at a depth of 1–1.5 cm below the water, making it invisible due to the opaque water. Testing trials were performed from two rotational starting positions, with the platform position fixed. Each trial ended as soon as the mouse reached the submerged platform and lasted up to 60 s if it failed to find the platform. After each testing trial, the mouse was allowed to remain on the platform for 120 s to learn the location of the platform and then returned to their cage. The escape latency and escape distance of each mouse were monitored by a SMART-LD program (Panlab, Barcelona, Spain). A quiet environment, consistent lighting, constant water temperature, and a fixed spatial frame were maintained throughout the experiment.

### 4.8. Probe Test

To quantitatively assess memory retention, a probe test was performed 24 h after the end of the water maze test. Testing took place with the platform removed from the pool used for the water maze test, and the mice were allowed to swim freely for 60 s. The swimming pattern of each mouse was monitored and recorded for 60 s using the SMART-LD program (Panlab, Barcelona, Spain). Spatial memory retention was quantified as the percentage of time spent in the target quadrant area.

### 4.9. Passive Avoidance Performance Test

The passive avoidance test is generally accepted as a simple method for testing memory. The passive avoidance test was performed in a step-through apparatus (Med Associates Inc., Fairfield, Vermont, USA) consisting of a light compartment and a dark compartment (each 20.3 × 15.9 × 21.3 cm) connected by a small gate. The floor of each compartment consisted of 3.175 mm stainless steel rods set 8 mm apart.

A training trial was performed 2 days after the probe test. In the training trial, the mouse was placed in the lighted compartment facing away from the dark compartment, and then when the mouse moved to a completely dark compartment, it received a 3 s electric shock (0.45 mA). The day after the training trial, mice were placed in the lighted compartment and the latency to enter the dark compartment was measured up to 3 min, which is referred to as step-through latency.

### 4.10. Collection and Preservation of Brain Tissues

After the completion of all the behavioral tests, the mice were perfused with PBS with heparin under inhaled CO_2_ anesthetization. The brain was immediately removed from the skull of the mouse and separated into the left and right brains. The isolated brain hemispheres were randomly stored on one side at −80 °C and the other side fixed in 10% formalin solution at room temperature for 3 days.

### 4.11. Thioflavin S Staining

The brain fixed in a 10% formalin solution was embedded in paraffin wax, and then the brain was cut into sections of 5 μm-thick slices. Thioflavin S staining was performed as described previously [52]. The sections were mounted in a mounting medium (Vectashield^®^ mounting medium for fluorescence with DAPI; Vector Laboratories, Burlingame, CA, USA). The thioflavin S-stained brain tissue image was obtained using a confocal fluorescence microscope (K1-Fluo; Nanoscope systems, Daejeon, Republic of Korea) (×50 and ×200).

### 4.12. Immunohistochemistry

The brain tissues were fixed in a 10% formalin solution and embedded in paraffin wax. Immunohistochemistry was performed as described previously [52]. To detect target proteins, specific antibodies against the glial fibrillary acidic protein (GFAP), ionized calcium-binding adaptor molecule 1 (IBA-1), inducible nitric oxide synthase (iNOS) (1:250; Abcam, Inc., Cambridge, MA, USA), and cyclooxygenase 2 (COX-2) (1:100, Novus Biologicals, Inc., Centennial, CO, USA) were used. Brain sections at 5 μm were visualized by target-specific antibodies with chromogen diaminobenzidine (Vector Laboratories, Burlingame, CA, USA), followed by mounting with Cytoseal XYL (Thermo Scientific, Waltham, MA, USA). Images of brain tissue were obtained using a light microscope (Microscope Axio Imager. A2; Carl Zeiss, Oberkochen, Germany; ×50 and ×200).

### 4.13. Western Blot Analysis

Brain hippocampus tissues were homogenized and lysed using a protein extraction solution (PRO-PREP, iNtRON, Kyungki-do, Republic of Korea). In the homogenized tissue, where the total protein concentration was determined using the Bradford reagent (Bio-Rad, Hercules, CA, USA), an amount equivalent to 40 µg of the total was separated by SDS/PAGE and transferred to Immobilon^®^ PVDF membranes (Millipore, Bedford, MA, USA). Membranes were blocked with 5% BSA in Tris-buffered saline containing 0.05% Tween-20 (TBST) for 1 h at room temperature and then incubated with specific primary antibodies in 2% BSA in TBST overnight at 4 °C. The membranes were triple-washed with TBST and incubated with diluted HRP-conjugated secondary antibodies in 2% BSA in TBST for 1 h at room temperature. After triple washing, the binding of antibodies to the PVDF membrane was detected using the Fusion FX 7 image acquisition system (Vilber Lourmat, Eberhardzell, Germany) with the Immobilon Western Chemiluminescent HRP Substrate (Millipore, Bedford, MA, USA). To detect target proteins, specific primary antibodies against iNOS, IBA-1, and GFAP (1:1000; Abcam, Inc., Cambridge, UK); COX-2 (1:1000; Novus Biologicals, Inc., CO, USA); ERK 1/2, p-ERK 1/2, nuclear factor-kappa B inhibitor alpha (IκBα), and p-IκBα (1:000; Cell Signaling Technology, Inc., Danvers, MA, USA); PTX3 (1:1000; Invitrogen, Waltham, MA, USA); and p-JNK and β-actin (1:200; Santa Cruz Biotechnology Inc., Santa Cruz, CA, USA) were used. The corresponding conjugated secondary antibodies such as anti-mouse, anti-rabbit, and anti-goat were purchased from Abcam.

### 4.14. Quantitative Real-Time PCR

The mRNA level was assessed by quantitative real-time polymerase chain reaction (qRT-PCR). Total RNA from hippocampus tissue was extracted using RiboEX (Geneall Biotechnology, Seoul, Republic of Korea). cDNA was synthesized using a High-Capacity cDNA Reverse Transcription kit (Thermo Scientific, Waltham, MA, USA). qRT-PCR was performed on a 7500 real-time PCR system (Applied Biosystems, Foster City, CA, USA) using HiPi Real-Time PCR SYBR green master mix (ELPIS biotech, Daejeon, Republic of Korea) with custom-designed primers. As a housekeeping control, β-actin was used. A total of 40 cycles were performed, consisting of an initial denaturation step of 3 min at 94 °C, a denaturation step of 30 s at 94 °C, an annealing step of 30 s at 56 °C, and an extension step of 1 min at 72 °C. The mRNA levels of target genes were normalized to β-actin. The primer pairs we used are shown in the Appendix A.

### 4.15. BV-2 Microglial Cell Culture and Transfection

Microglial BV-2 cells were obtained from the American Type Culture Collection (Rockville, MD, USA). BV-2 cells were maintained with serum-supplemented culture media of DMEM supplemented with FBS (10%) and antibiotics (100 units/mL). The BV-2 cells were incubated in the culture medium in a humidified incubator at 37 °C and 5% CO_2_. BV-2 cells were transiently transfected with siRNA (20 nM/well/6-well plate) or using the Lipofectamine^®^ RNAiMAX transfection reagent in Opti-MEM, according to the manufacturer’s specification (Invitrogen, Waltham, MA, USA). Negative control (NC) and CHI3L1 siRNA were purchased from OriGene Technologies, Inc. (Rockville, MD, USA).

### 4.16. Statistical Analysis

The data were statistically analyzed using the GraphPad Prism software (Version 4.03; GraphPad Software, Inc., San Diego, CA, USA). Data are presented as mean ± S.E.M. The group differences in all data were assessed by Student’s *t* test or one-way analysis of variance (ANOVA) followed by the Tukey multiple comparison test. A value of *p* < 0.05 was considered statistically significant. *, Significantly different between the two groups (*p* < 0.05); **, Significantly different between the two groups (*p* < 0.01); ***, Significantly different between the two groups (*p* < 0.001).

## Figures and Tables

**Figure 1 ijms-25-05550-f001:**
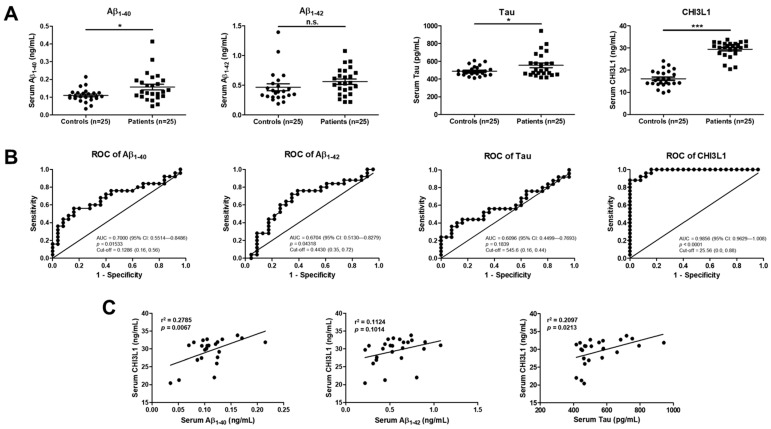
CHI3L1 related to AD. Human serum analysis of Aβ_1–40_, Aβ_1–42_, Tau, and CHI3L1 in patients with AD and healthy controls using ELISA (*n* = 25). The (**A**) serum levels and (**B**) ROC curves of Aβ_1–40_, Aβ_1–42_, Tau, and CHI3L1. (**C**) Spearman correlation test results between CHI3L1 and Aβ_1–40_, Aβ_1–42_, or Tau. Each value is mean ± S.E.M. from 25 samples. *, Significantly different from control group (*p* < 0.05). ***, Significantly different from control group (*p* < 0.001). n.s., not significant.

**Figure 2 ijms-25-05550-f002:**
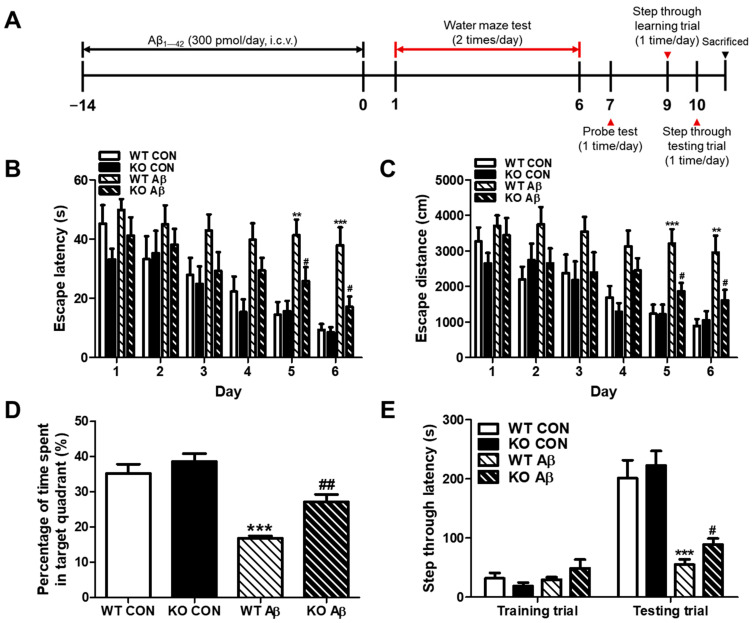
CHI3L1 deficiency improves cognitive function. (**A**) A timeline has been described that demonstrates the infusion of oligomeric Aβ and the assessment of cognitive function in CHI3L1 KO and WT mice. To investigate the effect of CHI3L1 on memory impairment, we carried out (**B**,**C**) the water maze test, (**D**) the probe test, and (**E**) the step-through type passive avoidance test. Memory and learning ability of mice were determined by the escape latencies (**B**, s) and escape distance (**C**, cm) for 6 days, and time spent in the target quadrant (**D**, %) in the probe test. Each value is mean ± S.E.M. from 8–10 mice. **, Significantly different from control WT group (*p* < 0.01). ***, Significantly different from control WT group (*p* < 0.001). #, Significantly different from Aβ-infused WT group (*p* < 0.05). ##, Significantly different from Aβ-infused WT group (*p* < 0.05).

**Figure 3 ijms-25-05550-f003:**
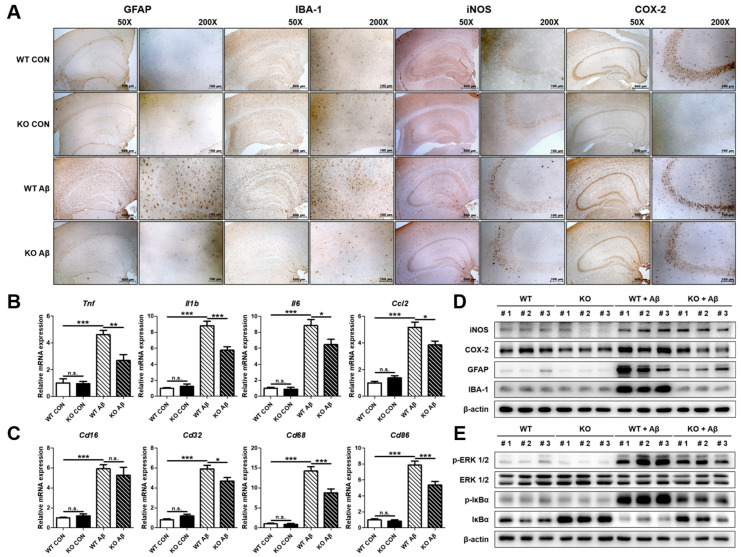
Effects of CHI3L1 deficiency on neuroinflammation in mice brain. (**A**) Expression of GFAP, IBA-1, iNOS, and COX-2 in Tg2576 mice hippocampus was determined by immunohistochemistry analysis. (**B**) The mRNA expression level of pro-inflammatory cytokines (*Tnf*, *Il1b*, and *Il6*) and chemokines (*Ccl2*) was assessed by qRT-PCR. (**C**) The mRNA expression level of the M1 microglia phenotype marker (*Cd16*, *Cd32*, *Cd68*, *Cd86*) was assessed by qRT-PCR. (**D**) Expression of iNOS, COX-2, GFAP, and IBA-1 was detected by Western blot (Appendix A). (**E**) Levels of p-ERK 1/2, ERK 1/2, p-IκBα, and IκBα were detected by Western blot. Each value is mean ± S.E.M. from 10–12 mice (Appendix A). *, Significantly different be-tween the two groups (*p* < 0.05); **, Significantly different between the two groups (*p* < 0.01); ***, Significantly different between the two groups (*p* < 0.001). n.s., not significant.

**Figure 4 ijms-25-05550-f004:**
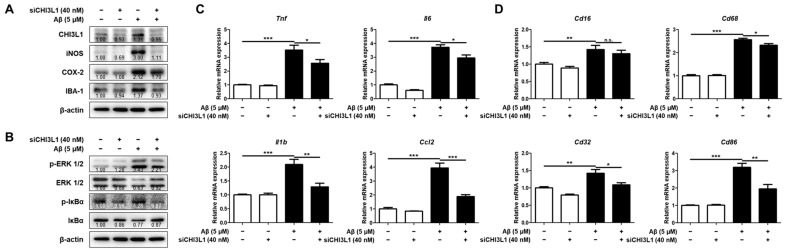
Effects of CHI3L1 deficiency on neuroinflammation in BV-2 cells. BV-2 cells were transfected with CHI3L1 siRNA (40 nM). After 24 h, cells were treated with Aβ (5 µM) for 24 h. (**A**) Expression of iNOS, COX-2, and IBA-1 was detected by Western blot. (**B**) Levels of p-ERK 1/2, ERK 1/2, p-IκBα, and IκBα were detected by Western blot. (**C**) The mRNA expression level of pro-inflammatory cytokines (*Tnf*, *Il1b*, and *Il6*) and chemokines (*Ccl2*) was assessed by qRT-PCR. (**D**) The mRNA expression level of the M1 microglia phenotype marker (*Cd16*, *Cd32*, *Cd68*, *Cd86*) was assessed by qRT-PCR. Each value is mean ± S.E.M. from 10–12 samples. *, Significantly different be-tween the two groups (*p* < 0.05); **, Significantly different between the two groups (*p* < 0.01); ***, Significantly different between the two groups (*p* < 0.001). n.s., not significant.

**Figure 5 ijms-25-05550-f005:**
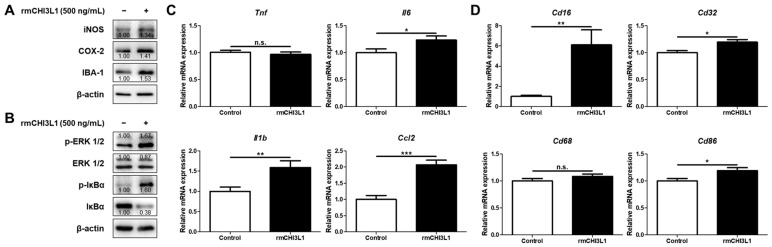
Effect of CHI3L1 on neuroinflammation in BV-2 cells. BV-2 cells were treated with rmCHI3L1 (500 ng/mL) for 24 h. (**A**) Expression of iNOS, COX-2, and IBA-1 was detected by Western blot. (**B**) Levels of p-ERK 1/2, ERK 1/2, p-IκBα, and IκBα were detected by Western blot. (**C**) The mRNA expression level of pro-inflammatory cytokines (*Tnf*, *Il1b*, and *Il6*) and chemokines (*Ccl2*) was assessed by qRT-PCR. (**D**) The mRNA expression level of the M1 microglia phenotype marker (*Cd16*, *Cd32*, *Cd68*, *Cd86*) was assessed by qRT-PCR. Each value is mean ± S.E.M. from 11–12 samples. *, Significantly different be-tween the two groups (*p* < 0.05); **, Significantly different between the two groups (*p* < 0.01); ***, Significantly different between the two groups (*p* < 0.001). n.s., not significant.

**Figure 6 ijms-25-05550-f006:**
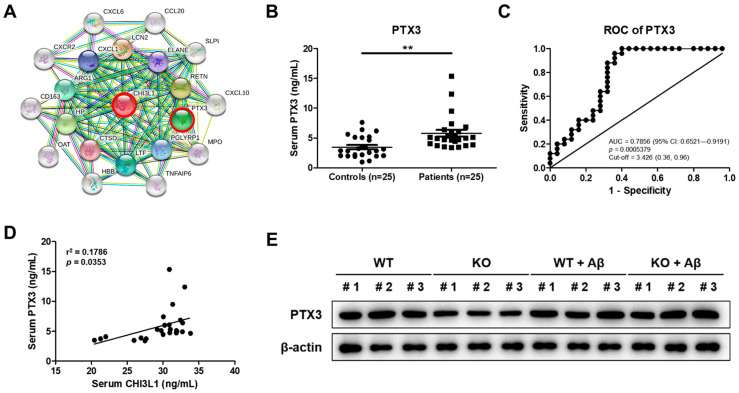
PTX3 is associated with CHI3L1 as well as AD. (**A**) Gene network analysis associated with CHI3L1 was carried out using the web-based analysis tool. Serum analysis of PTX3 in patients with AD. (**B**) The serum levels and (**C**) ROC curve of PTX3 in patients with AD and healthy controls. (**D**) Spearman correlation test results between PTX3 and CHI3L1. (**E**) Expression of PTX3 in the mouse brain was detected by Western blot. Each value is mean ± S.E.M. from 25 samples (Appendix A). **, Significantly different from control group (*p* < 0.01).

## Data Availability

The datasets generated and/or analyzed during the current study are available from the corresponding author upon reasonable request.

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
