# Peer review of "Inhibition of Amyloid-β (Aβ)-Induced Cognitive Impairment and Neuroinflammation in CHI3L1 Knockout Mice through Downregulation of ERK-PTX3 Pathway"

_ijms, 2024, doi:10.3390/ijms25105550_

Round 1

Reviewer 1 Report

Comments and Suggestions for Authors

The reviewer had some concerns in the present manuscript.

Abstract; correct a spelling error "inhibitinf", and spell out an abbreviation "ROC".

Methods, Preparation of oligomeric Abeta; how was the extent of oligomerization after incubation at 4℃ for 12 h? The day before surgery, Abeta was diluted to 50 microM and infused into the mouse brain for 14 days. During that time, was Abeta solution in a pump at room temperature?

Methods, 2.8. Probe test; was probe test an assessment in Morris Water Maze test?

Methods, 2.15.; in the experiments using BV-2 cells, what was the experimental condition including preparation of Abeta, time course of transfection and Abeta treatments, the explanation of rmCHI3L1, and so on?

Line 302, "(C)" was lost.

Results 3.2. and Figure 2D; if Abeta impaired memory, why was not the percentage of time spent in target quadrant 25% (a quarter)?

Results 3.3.; Supplementary figures were not contained in the review version, and so the reviewer could not judge.

Results 3.4. and Figure 3A, 3D and 3E; the quantitative graphs should be added if the authors wanted to say 'increase' and 'decrease'. For example, COX-2 in Figure 3A, the expression was high in WT CON and low in KO CON, and Abeta looked not to increase the expression of COX-2 between WT CON and WT Abeta. 

Similarly, Figure 4A and 4B, Figure 5A and 5B, and Figure 6E; the quantitative graphs should be added.

In the present study, how about the changes of anti-inflammatory factors and M2 microglial markers?

Reviewer 2 Report

Comments and Suggestions for Authors

Inhibition of amyloid-β (Aβ)-induced cognitive impairment and neuroinflammation in CHI3L1 knockout mice through downregulation of ERK-PTX3 pathway

Please paraphrase your methods since they currently show a large amount of similarity to other sources. This is very important. The authors are good scientists and write well, so there is no reason there should be this level of similarity with another text. 

The abstract starts with introducing the role of CHI3L1 across diseases and the role of this gene in AD, they also refer to the inhibitor that they developed against this in a previous study which improved some functions in mouse AD models. So, they aim to test the loss of CHI3L1 in mice and cell lines but seems to have a positive effect. The authors employ some web-based tools as well and through this study they discover a target for CHI3L1, and PTX3 and make some further inquiries.

The abstract is clear enough but the authors can add some conclusions and implications to the end of the abstract.

Intro: The authors introduce AD and cover some of its pathophysiology. The topic continues with inflammatory states of astrocytes and microglia that may affect neurons. The topic continues with content on M1 and M2 microglial cells which are in many ways similar to M1/M2 macrophages.

Then as expected the authors move on to introducing CHI3L1, the properties and the cell types which express them also get covered in good depth. This is then nicely linked to an inflammatory phenotype in AD. 

It is quite interesting that CHI3L1 can influence important inflammatory mediators such as TNF and Il1B.

In the last paragraph of the intro, the authors use a much larger font, please kindly correct this.

The authors formulated their aims.

Overall, the intro was relevant and useful.

Methods:

The authors describe the patient tissue that they have obtained.

This is followed by cat numbers for reagents used in different assays.

The behavioural tests are described in good depth.

Overall, with more paraphrasing, the methods section is pretty good.

Results:

Figure 1: the authors measure AB and tau proteins in AD patients using ELISA. This was followed by measuring RO-AUC, cut-off value, specificity and sensitivity to the 4 markers. Spemann’s test also showed a correlation between CHI3L1 and AB1-4. Which is great. The figure legend of Figure 1 is brief, please explain all subunits of the figure.

Figure 2: escape latency was increased for the AB infusion group but for the CHI3L1KO mice receiving the infusion, this was decreased. This suggested that the absence of CHI3L1 improves AB-induced memory loss.

In Figure 2, the scheme of experiment planning is really useful.

The font keeps changing please rectify it.

Supplementary data for ELISA, unfortunately, this data was not uploaded so I was not able to evaluate it. Based on the main text, the authors outline the decreased levels of AB1-40 in the KO mice. This is an interesting result and it would be good to get a mechanism by which CHI3L1 is reducing brain section AB deposits.

The authors should also add methods for performing ELISA on these sections since the ELISA section does not specifically explain it.

Figure 3 and of course, the expression profile of KO mice tissue infused with AB. Initially, tissue section IHC for iNOS Cox2 and these were decreased in CHI3L1 KO mice. ERK and NF-kB also get tested on protein levels. On mRNA levels, a platform of inflammatory genes gets tested. Of course, compared to no infusion, wt infused and KO infused are both higher but within the infused category, the KO infused consistently shows lower inflammatory markers. These are really good results.

Please expand on figure legends. Also, for all the western blots, please provide a bar chart of their quantification and if relevant statistical analysis.

Figure 4: The authors move to CHI3L1 deficient cell lines to corroborate the readouts in Figure 3. The data looks good.

Figure 5: The authors move to CHI3L1 expressing cell lines to test the opposite scenario. 

Figure 6: CHI3L1 correlated with PTX3, the WB quantification and comparison is especially relevant to 6E. At the start of the study, I thought that perhaps PTX3 might be a target of CHI3L1 and based on the 6E WB KO data (1-3) it seems that it might slightly be downregulated.

The study has nicely evidenced the various readouts (behavioural tests, changes to mRNA and protein levels of inflammatory markers in vivo and cell lines). One aspect that is remaining is the mechanism and I was hoping that the PTX3 link might shed more light on the mechanism.

Aspects that might be useful to substantiate:

1- The mechanism by which CHI3L1 is reducing brain section AB deposits (reducing them in CHI3L1 KOs (based on your ELISA test). Is this due to the increase in inflammatory mediators that you have shown in Figure 5?

2- Is there a middleman between CHI3L1 and inflammatory mediators and genes? Presumably, there must be several middlemen between a glycoprotein and triggering gene expression in the nucleus. Is one of these PTX3? Does the knockdown of PTX3 mirror the effects of CHI3L1 KO? If so, it is a target. I suspect the authors might have some evidence about PTX3 but might be keeping this for another paper. At any rate, the PTX3 aspect could be substantiated.

The data in 6A shows several other mediators that could be directly linked to PTX3. One interesting aspect can be CXCR2, CCL20 and CXCL1 and these are worth investigating to better understand the “stream of events”.

3- What is the missing link between improving cognition and memory and the absence of CHI3L1 in these mice? Does the inflammatory profile explain this link? What is the role of the AB plaques?

4- The authors do allude to a potential chain of events in the discussion and that is CHI3L1 binding to RAGE and then triggering NF-kB, ERK and STAT3. This in turn activated IL-6, TNF and IL-6. This can indeed be 

Discussion, evidence from the literature is offered to support the findings of this study.

It starts with CHI3L1 as a biomarker for AD patients and given this data it seems that it could be an important marker to look at.

KO of CHI3L1 improves cognition in the author’s current and previous study and it would be good to get a mechanistic insight.

A lot of emphasis is given to inflammatory profiles in AD.

The last paragraph is about PTX3 but it falls short of what it does (in association with CHI3L1).

The discussion could have a limitations section.

The font of the refs is different to the main text.

Overall, this was a well-thought-out and comprehensive study but there are areas of it that can be improved.

Comments on the Quality of English Language

Some editing required

Reviewer 3 Report

Comments and Suggestions for Authors

The "Inhibition of amyloid-β (Aβ)-induced cognitive impairment and neuroinflammation in CHI3L1 knockout mice through downregulation of ERK-PTX3 pathway " manuscript shows important advances on the relationship of Chitinase-3-like 1 (CHI3L1) and AD. However, the authors could improve the research by providing information:
 -The measurement of IL-1β and IL-6 that positively regulate CHI3L1.
- The authors found a relationship with NF-kB, however, it would be important at what level it exerts its effect (IκB kinases, IKKα and IKKβ).
- For better understanding change the name of the compound 2-({3-[2-(1-Cyclohexen-1-yl)ethyl]-6,7-dimethoxy-4-oxo-3,4-dihydro-2-quinazolinyl} sulfanyl)-N-(4-ethylphenyl)butanamide by K284-6111. Also, include the origin of the reagent.
- Because in vivo cytotoxicity or toxicity studies of compound K284-6111 were not included.

Round 2

Reviewer 1 Report

Comments and Suggestions for Authors

The authors revised the manuscript and responded the reviewer's comments.

Author Response

Thank for comment. As you pointed out, we described the materials and methods section more detail as shown in the revised MS.

Reviewer 2 Report

Comments and Suggestions for Authors

The authors have tried to reasonably respond to my comment.

One point that I cannot establish: whether the method's text similarity has been addressed (paraphrasing)

Also, please check if the plots need to have lane names (these are the original blots I am referring to)

If these have been addressed, then I have no further comments.

Author Response

Thank for comment. As you pointed out, we paraphrased the Materials and Methods sections as shown in the revised MS. We have also attached the missing original western blot image.